# Complete Genome and Molecular Characterization of a New Cyprinid Herpesvirus 2 (CyHV-2) SH-01 Strain Isolated from Cultured Crucian Carp

**DOI:** 10.3390/v14092068

**Published:** 2022-09-17

**Authors:** Jia Yang, Jinxuan Wen, Simin Xiao, Chang Wei, Fei Yu, Patarida Roengjit, Liqun Lu, Hao Wang

**Affiliations:** 1National Pathogen Collection Center for Aquatic Animals, Shanghai Ocean University, Shanghai 201306, China; 2National Demonstration Center for Experimental Fisheries Science Education, Shanghai Ocean University, Shanghai 201306, China; 3Department of Ocean Technology, College of Marine and Bioengineering, Yancheng Institute of Technology, Yancheng 224051, China; 4Institute of Marine Biology, College of Oceanography, Hohai University, Nanjing 210098, China; 5Faculty of Agricultural Technology, Phuket Rajabhat University, Phuket 83000, Thailand; 6Key Laboratory of Freshwater Aquatic Genetic Resources, Ministry of Agriculture, Shanghai Ocean University, Shanghai 201306, China

**Keywords:** cyprinid herpesvirus 2, CyHV-2, herpesviral hematopoietic necrosis, HVHN, crucian carp, *Carassius carassius*, complete genome, molecular characterization, genome collinearity

## Abstract

Cyprinid herpesvirus 2 (CyHV-2) is a causative factor of herpesviral hematopoietic necrosis (HVHN) in farmed crucian carp (*Carassius carassius*) and goldfish (*Carassius auratus*). In this study, we analyzed the genomic characteristics of a new strain, CyHV-2 SH-01, isolated during outbreaks in crucian carp at a local fish farm near Shanghai, China. CyHV-2 SH-01 exhibited a high sensitivity to goldfish and crucian carp in our previous research. The complete genome of SH-01 is 290,428 bp with 154 potential open reading frames (ORFs) and terminal repeat (TR) regions at both ends. Compared to the sequenced genomes of other CyHVs, *Carassius auratus* herpesvirus (CaHV) and Anguillid herpesvirus 1 (AngHV-1), several variations were found in SH-01, including nucleotide mutations, deletions, and insertions, as well as gene duplications, rearrangements, and horizontal transfers. Overall, the genome of SH-01 shares 99.60% of its identity with that of ST-J1. Genomic collinearity analysis showed that SH-01 has a high degree of collinearity with another three CyHV-2 isolates, and it is generally closely related to CaHV, CyHV-1, and CyHV-3, although it contains many differences in locally collinear blocks (LCBs). The lowest degree of collinearity was found with AngHV-1, despite some homologous LCBs, indicating that they are evolutionarily the most distantly related. The results provide new clues to better understand the CyHV-2 genome through sequencing and sequence mining.

## 1. Introduction

Crucian carp (*Carassius carassius*) is one of the most widely farmed freshwater fish species of the *Cyprinidae* family in China, alongside goldfish (*Carassius auratus*), and the annual worldwide production reached 2748.6 thousand tonnes in 2020 (FAO, 2022). Viral diseases are common in aquaculture, including the crucian carp farming industry, and they cause serious harm to wild lower vertebrate populations worldwide.

The cyprinid herpesvirus 2 (CyHV-2) infection was originally identified as herpesviral hematopoietic necrosis (HVHN) in ornamental goldfish, associated with high mortality in Japan in 1995 [1]. Subsequently, the virus has rapidly spread to many countries and regions worldwide, including Australia [2], the USA [3,4], the UK [5], Hungary [6], Czech Republic [7], China [8], Italy [9], France [10], the Netherlands [11], India [12], Switzerland [13], Germany [14], Turkey [15], and Poland [16]. CyHV-2 causes HVHN in crucian carp and goldfish with high susceptibility and mortality. It is a double-stranded DNA virus, and a member of the genus *Cyprinivirus* of the family *Alloherpesviridae*, in which four species, namely CyHV-1, CyHV-2, CyHV-3, and Anguillid herpesvirus 1 (AngHV-1), have hitherto been classified into the genus (ICTV, 2021). Similar to other herpesviruses, CyHVs also exhibit latency and long-term persistence, which can be reactivated under some stresses such as changes in water temperature [17,18,19,20,21], representing a huge threat to carp. Phylogenetically, the three CyHVs are closely associated [22], with CyHV-2 and CyHV-3 more closely related to each other than to CyHV-1, while AngHV-1 is the next most closely related to the CyHVs, and other alloherpesviruses are much more distantly related [23,24,25]. Additionally, a new *Carassius auratus* herpesvirus (CaHV) strain that the classification status has not been clarified according to the ICTV (2021), was reported that caused acute gill hemorrhages and high mortality in crucian carp, and it was closely related to known CyHVs despite several differences in the genome structures [26].

Research in the past decade has mainly involved detection and genome sequencing of CyHV-2. To date, six CyHV-2 isolates have been cultivated and genome sequenced, including ST-J1 (GenBank Accession No. NC_019495), SY-C1 (KM200722), SY (KT387800), CNDF-TB2015 (MN201961), YZ-01 (MK260012), and YC-01 (MN593216). Of these CyHV-2 isolates, ST-J1 was isolated from goldfish, and the other five isolates were from crucian carp. It is important to further explore the genome structure and potential molecular pathogenic mechanisms of CyHV-2 using complete genome sequencing, comparative genomics, and molecular characterization. Although CyHV-2 is distributed worldwide, different CyHV-2 isolates have not been comprehensively compared, especially recently discovered isolates.

Previously, we isolated a new strain, CyHV-2 SH-01, during outbreaks in crucian carp at a local fish farm near Shanghai, China and confirmed that goldfish also showed high susceptibility with symptoms including acute gill hemorrhages and high mortality, similar to HVHN caused by SH-01 isolated from crucian carp [27]. To further explore the genetic properties and potential molecular pathogenic mechanisms of CyHV-2, we sequenced and analyzed the genome of CyHV-2 SH-01 in the present work.

## 2. Materials and Methods

### 2.1. Isolation of Virus and DNA Extraction

In a previous work in our laboratory (Shanghai Ocean University), moribund crucian carp (13–15 cm in length) were collected during a disease outbreak in a fish farm near Shanghai, China, and diseased tissues including kidney, spleen, muscle, and blood were collected for testing and were identified as HVHN caused by CyHV-2, and the strain named SH-01 was isolated. Then, DNA was extracted from purified viral particles using the TIANamp Genomic DNA Kit (DP304, Tiangen, Beijing, China) according to the manufacturer’s instructions, and PCR amplification was performed using the PrimeSTAR^®^ Max system (R045Q, TaKaRa, Beijing, China) for sequencing, as described previously reported [27].

### 2.2. DNA Sequencing, Genome Assembly, and Annotation

The genome sequencing of CyHV-2 SH-01 was commercially performed by the NextOmics company (Wuhan, China) using a whole-genome shotgun (WGS) strategy through second-generation sequencing. Briefly, the DNA extracted from samples were followed by evaluation for quality control. Then, the quality-checked DNA samples were prepared to construct a second-generation DNA library of small fragments based on the characteristics of the genome, and WGS data were obtained by paired-end sequencing on an Illumina Hiseq Xten platform (Illumina, San Diego, CA, USA). The raw data contained some low-quality readings with adapters and were filtered using fastp v0.20.0 (https://github.com/OpenGene/fastp accessed on 19 June 2022) to obtain clean data followed by fastqc (https://github.com/s-andrews/FastQC accessed on 19 June 2022) for quality control. Then, the quality-controlled data were compared with the virus database in Refseq (https://www.ncbi.nlm.nih.gov/refseq/ accessed on 19 June 2022); 50 Mb readings with successful comparisons were extracted for de novo assembly using unicycler v0.4.8 [28] (https://github.com/rrwick/Unicycler accessed on 19 June 2022; parameters: -1 r1 -2 r2 --keep3 --mode normal), and then the genome was corrected and optimised by pilon v1.23 [29] (https://github.com/broadinstitute/pilon accessed on 19 June 2022; parameters: default). Finally, one contig was obtained and was aligned with the reference genome sequences of ST-J1, YZ-01, and SY-C1. All DNA sequences were then compared with all known protein-encoding sequences from other published reference strains using BLAST [30] (https://blast.ncbi.nlm.nih.gov/Blast.cgi accessed on 19 June 2022), and coding genes were predicted using prokka v1.14.6 [31] (https://github.com/tseemann/prokka accessed on 19 June 2022; parameters: --kingdom Viruses --gcode 1), yielding complete coding sequences (CDS).

### 2.3. Analysis of the Genome Structure and Molecular Characterization of CyHV-2 SH-01

The frequency of codon usage in the SH-01 genome was analyzed using the CUSP program (https://www.bioinformatics.nl/cgi-bin/emboss/cusp accessed on 17 June 2022). The genome map of SH-01 was drawn with Adobe Illustrator 2021 (Adobe, San Jose, CA, USA). A graph of the sequence lengths of the amino acids of proteins encoded by open reading frames (ORFs) on the X-axis and the number of proteins per length on the Y-axis was calculated by Geneious Prime v2022.2.1 (Biomatters, Auckland, New Zealand).

The signal peptide (SP) sequences and transmembrane domains (TMDs) of proteins encoded by SH-01 were predicted using signalP-5.0 (https://services.healthtech.dtu.dk/service.php?SignalP-5.0 accessed on 21 June 2022) and TMHMM 2.0 (https://services.healthtech.dtu.dk/service.php?TMHMM-2.0 accessed on 21 June 2022), respectively. Then, the function features of all the proteins encoded by SH-01 were also predicted and analyzed using the conserved domains database (CDD; https://www.ncbi.nlm.nih.gov/cdd accessed on 21 June 2022) with an E-value < e^−5^ as a cut-off [32].

### 2.4. Comparison of Genomic Structure and Evolutionary Relationships among SH-01 and the Other Seven Strains

The sequence identities of the genomes and ORFs (or CDS) among SH-01 and another six closely related strains in the genus *Cyprinivirus* and CaHV (the classification status has not been clarified according to the ICTV 2021) were aligned through MAFFT by Geneious Prime v2022.2.1 (Biomatters, Auckland, New Zealand).

The evolutionary patterns among the homologous or heterologous regions of the genomes of eight isolates including SH-01 were analyzed by Mauve alignment in DNASTAR Lasergene v17.3 (DNASTAR, Madison, WI, USA). Furthermore, the comparison of locally collinear blocks (LCBs) among SH-01, CyHV-1, and CyHV-3 was conducted using the progressive Mauve algorithm in Geneious Prime v2022.2.1 (Biomatters, Auckland, New Zealand). Then, a phylogenetic tree was constructed based on the amino acid sequences of helicase (ORF71) using the neighbor-joining method in MEGA v11 (https://megasoftware.net accessed on 10 July 2022) with bootstrap values of 1000 replications.

## 3. Results

### 3.1. Genome Structure and Composition

We obtained a total of 46,723,798 raw readings (7,008,569,700 raw bases) by high-throughput sequencing (HTS), and after removing low-quality data, 46,357,846 clean readings (6,461,792,984 clean bases) remained, with a G+C content of 59.50%. Then, we successfully assembled the complete genome sequence of CyHV-2 SH-01 and submitted it to GenBank (Accession No. BankIt2436221).

The genome of CyHV-2 SH-01 is a linear double-stranded DNA, 290,428 bp in length, with an overall G+C content of around 51.60%, including a unique (U) and terminal repeat (TR) region at both ends. We analyzed the frequency of codon usage in the SH-01 genome using the CUSP program, and the results showed that the coding GC content of SH-01 open reading frames (ORFs) was 51.64%, while the GC content of 1st, 2nd, and 3rd letters in the triplet codons were 52.61%, 52.38%, and 49.93%, respectively. The SH-01 genome contains 154 predicted ORFs, of which four duplicated ORFs (*ORF5*, *ORF6*, *ORF7*, and *ORF8*) are located in TR (Figure 1A), similar to ST-J1 [33]. ORFs encode proteins ranging in length from 63 (ORF106) to 4123 (ORF62) amino acids (aa), with an average length of 527.37 aa (Figure 1B and Appendix A). Thirty-two ORFs contain introns, of which *ORF79* has four introns, *ORF6*, *ORF33*, and *ORF3* have three introns, and the other 28 ORFs have one or two introns (Appendix A). In line with ST-J1 and SY, there are 86 ORFs located on the positive strand in SH-01 and 68 ORFs on the negative strand. Additionally, there are seven core ORFs (*ORF19*, *ORF55*, *ORF72*, *ORF88*, *ORF92*, *ORF93*, and *ORF142*) in the SH-01 genome (Figure 1A and Appendix A), of which *ORF72* and *ORF92* are significantly conserved among alloherpesviruses.

### 3.2. Chronological Characteristics of Gene Expression

Gene expression during lytic replication of herpesviruses is characterized by a distinct chronological sequence involving three main temporal phases, immediate-early (IE), early (E), and late (L) genes, and expression patterns are the result of complex interactions between viruses and cytokines at the transcriptional and post-transcriptional levels, as well as structural differences in the promoters (*cis*- versus *trans*-acting elements) among the three types of genes [34,35]. Similar to herpesviruses, with reference to CyHV-2 ST-J1 [36], we marked the five IE (red), 34 E (green), and 39 L (blue) genes in the genome map of SH-01 (Figure 1A and Appendix A). Further understanding of the chronological characteristics of the gene expression of CyHV-2 could provide insight into the viral replication mechanisms and interactions with hosts.

### 3.3. Features of the Predicted Functional Protein Encoded by SH-01

Among the 154 ORFs of CyHV-2 SH-01, 26 ORFs encoding proteins were predicted to possess a SP sequence, and seventy-five putative proteins were predicted to possess one or more TMDs. All 154 predicted proteins were also analyzed through the CDD database, and 55 putative proteins were predicted to contain one or more conserved domains (Appendix A). These results revealed that six of the ORFs (*ORF25C*, *ORF34*, *ORF52*, *ORF119*, *ORF127*, and *ORF151A*) encoding proteins contain an SP but no TMD, suggesting that these proteins are secreted. Notably, ORF64, ORF114, ORF152A, ORF16, and ORF153B have 10, 9, 8, 7, and 6 TMDs, respectively, and these were predicted to be important membrane proteins of SH-01, similar to CyHV-3 [33]. As shown in Appendix A, there are 10 putative TMDs in ORF64, similar to the 12 TMDs in ORF64 of CyHV-3, indicating a nucleoside transporter domain or similar, suggesting that the protein may be essential for CyHV-2 replication, but this needs to be verified experimentally. In addition, there are three ORFs (ORF41, ORF144, and ORF150) with a really interesting new gene (RING) domain that have a ubiquitin and possess protein transferase activity, one ORF (ORF4) belonging to the tumor necrosis factor receptor (TNFR) family, and 143 unclassified ORFs (Figure 1A and Appendix A).

Furthermore, *ORF71* encodes a DEAD (Asp-Glu-Ala-Asp)-like helicase–primase subunit, in which the N-terminal domain contains an ATP-binding region involved in ATP-dependent RNA or DNA unwinding, and *ORF79* encodes a putative DNA polymerase catalytic subunit, which functions in viral replication. Additionally, *ORF19* and *ORF140* encode nucleotide kinases; *ORF122* and *ORF123* encode trimeric dUTP diphosphatases that function in the preservation of chromosomal integrity; ORF51, ORF68, and ORF91 possess structural maintenance of chromosomes (SMC) domains; *ORF141* and *ORF142* encode ribonucleotide reductases; and *ORF52* and *ORF55* encode a ribonuclease and thymidine kinase, respectively, which are involved in DNA degradation and provide a nucleotide feedstock for viral DNA synthesis. *ORF68* encodes SbcC [37], part of an exonuclease complex involved in DNA metabolism, replication, recombination, and damage repair, as well as signal transduction and immune responses, and there is a macrodomain in ORF68 and ORF69 with the same function. *ORF128* encodes rad18 (DNA repair protein) that plays a role in DNA metabolism, replication, recombination, and repair. ORF129 has a putative SPRY (B30.2) domain, part of tripartite motif 5α (TRIM5α), considered to be a specific anti-retroviral determinant [38]. ORF145 has an RNA recognition motif (RRM) involved in post-transcriptional gene expression processes including mRNA and rRNA processing, RNA export, and RNA stability [39]. *ORF28A*, *ORF146*, and *ORF147* encode proteins with immunoglobulin (Ig) domains and are the members of the Ig superfamily. *ORF104* encodes a protein kinase (PKc-like) associated with protein phosphorylation and regulation of many cellular signaling pathways. ORF20 and ORF28 contain an NAD(P)(+)-binding (NADB) Rossmann-fold domain related to redox metabolic pathways. *ORF98* encodes an uracil-DNA glycosylase (UDG)-like protein, which initiates the repair of uracil bases in DNA and maintains the integrity of genetic information. Moreover, ORF62 contains an ovarian tumor (OTU) domain, which may be involved in the suppression of the innate immune responses of hosts [40].

In addition, ORF99 is predicted to be the same as in spike-torovirus, a transmembrane protein of coronaviruses that mediates the binding of viruses to host cell receptors and participates in membrane fusion. *ORF139* encodes a protein homologous to the C-terminal structure of the poxvirus B22R protein. Furthermore, *ORF155* encodes a protein homologous to the major outer-envelope glycoprotein (BLLF1) of EBV, also known as gp350, which is abundantly expressed in the envelope of EBV and is the antigen responsible for stimulating neutralizing antibody production in host cells [41]. *ORF30* encodes a protein homologous to the late lytic protein BDLF3 of EBV associated with immune evasion [42]. The protein encoded by *ORF49* is homologous to infected-cell polypeptide 4 (ICP4), a major transcriptional regulator of the herpes simplex virus type 1 (HSV-1), forming a tripartite complex with the TATA-binding protein (TBP) and the transcription factor IIB (TFIIB), related to the activation of L genes [35,43].

### 3.4. Comparison of Genome Structure and ORFs Arrangement

The genomes of three isolates of CyHV-2, namely ST-J1, SY-C1, and SY, have been sequenced and annotated through comparative genomics [33,44,45]. They are 290,304, 289,365, and 290,455 bp in length, encoding 150, 140, and 150 unique ORFs, respectively, and all genomes possess U and TR features at each end. Consistently, our results showed that the genome of SH-01 is 290,428 bp in length, with U and TR at each end, encoding 150 unique ORFs, sharing 99.60%, 98.53%, and 98.35% sequence identity with ST-J1, SY-C1, and SY, respectively (Table 1). However, the genome of SH-01 shares the greatest similarity with ST-J1, and SY-C1 lacks 11 ORFs present in the other three strains, of which one (*ORF7*) is in the TR and the other 10 are in the U features, while *ORF28* and *ORF28A* in SY-C1 are reversed, corresponding to *ORF28A* and *ORF28* in the other three strains, respectively, based on the comparison with SH-01 (Figure 2). Compared with the whole genome sequence of ST-J1, SH-01 has 116 nucleotide deletion sites (716 bp in total) and 82 nucleotide insertion sites (1073 bp in total). However, 71 out of the 150 ORFs are identical (100%) between SH-01 and ST-J1, and only 23 (of the total 154 ORFs) have no variation in SH-01 compared with ST-J1, SY, and SY-C1 (Appendix A).

Compared with CyHV-1 and CyHV-3, similar to ST-J1, the counterparts of some ORFs in the TR in CyHV-3 (*ORF1*, *ORF2*, *ORF3*, and *ORF4*) or CyHV-1 (*ORF2* and *ORF3*) [33] are located closely downstream of the U in SH-01, related to flanking genes, with only *ORF5*, *ORF6*, *ORF7*, and *ORF8* in the TR of SH-01. Moreover, the CyHV-3 genome remains the largest overall, and CyHV-1 contains the fewest ORFs among CyHVs, but unlike CyHV-1 and CyHV-3, the CyHV-2 SH-01 genome has more complexity in terms of copy size and arrangement of the U and the TR. Similar to ST-J1, SH-01 also contains a 220 bp inverted repeat region downstream of *ORF25C* and *ORF48*, but it was not observed in SY, SY-C1, or CaHV [26], nor in CyHV-1 and CyHV-3. Compared with the genomic structure of CyHV-3, *ORF4* is located downstream of the U rather than in the TR in SH-01, while *ORF4* is deleted in CyHV-1. Importantly, *ORF140* undergoes a large translocation in CyHV-1 and is located upstream of the U, while in the CyHV-2 and CyHV-3 strains it is located downstream. In addition, *ORF2A* is inserted between *ORF2* and *ORF3* in SH-01, and there are many insertions, deletions, rearrangements, and inversions in the arrangement of ORFs in SH-01; for example, the orientation of *ORF128*–*133* and *ORF135*–*138* is opposite compared to CyHV-1 and CyHV-3 (Figure 2 and Appendix A).

Moreover, the genome of SH-01 also shares a high identity (92.63%) with CaHV [26], although CaHV has no TR repeats, and *ORF1*–*13* at the downstream end of its genome correspond to *ORF153B*–*ORF8* at the upstream end of the SH-01 genome, while *ORF144* in CaHV corresponds to *ORF4* in SH-01 (Table 1 and Figure 2). Additionally, among the genus *Cyprinivirus*, AngHV-1 is distantly related to SH-01, with only 53 homologous ORFs, all with less than 60% identity; for example, *ORF5* in the TR of AngHV-1 is homologous to *ORF123* in the U of SH-01 (Appendix A and Figure 2).

### 3.5. Genomic Evolutionary Relationships among SH-01 and the Other Seven Strains

Based on the sequencing and analysis of viruses of the genus *Cyprinivirus*, the genomes of CyHVs have been in found multiple genetic information changes such as gene recombination, including deletions, duplications, rearrangements, and horizontal transfers, or nucleotide mutations, including base substitutions, insertions, and deletions, which has resulted in a high complexity and diversity of these genomes through evolution. Typically, orthologous genes can “jump” within genomes, known as gene rearrangement, and maximal collinear sets of homologous sites are regarded as locally collinear blocks (LCBs) that cover a “block” of sequences without any internal genome rearrangement, or causing changes in genome structure by inserting as “new genes” into other genomes [47].

Herein, we analyzed the evolutionary patterns among the homologous or heterologous regions of the genomes of eight isolates, including SH-01. The evolutionary relationships between the genomes of SH-01 and the other seven isolates were explored (Figure 3A). Different LCBs are marked with regions in different colors, and these can “jump” or rearrange across the genome as a complete unit. In contrast to SH-01, LCBs on the same side of the centerline indicate the same transcriptional direction, while those on the opposite side indicate an inverted transcriptional direction. The spacer region (outside the LCBs) is considered to have no or very low homology, and the colored part inside the LCBs shows the corresponding parts of the homologous gene sequences (Figure 3A,B).

The genome of SH-01 contains 19 LCBs with variation in sequence length and in at least one ORF inside each LCB. SH-01 showed high consistency in orientation and alignment of LCBs compared with the other three CyHV-2 isolates, despite differences in the number of LCBs. In particular, the corresponding lengths of the same LCBs were also highly similar, suggesting that these four isolates may have similar pathways in genome evolution. Consistent with the results in Figure 2, LCB14–18 (downstream end of the genome) of SH-01 correspond to LCB1–8 (upstream end of the genome) of CaHV. In contrast, compared with CyHV-1, CyHV-3, and AngHV-1, there are many differences in the number, orientation, alignment order, and corresponding length of the LCBs. Specifically, compared with CyHVs, AngHV-1 has the most divergent LCBs, and only seven are homologous. Notably, consistent with Figure 2, *ORF140* jumped to LCB4 (upstream end of the genome) in CyHV-1, while it is located at LCB7, LCB22, and LCB15 (downstream end of the genome) in CyHV-2, CyHV-3, and CaHV, respectively. In addition, there are many inversions of the LCBs in CyHV-1 and CyHV-3, as shown in Figure 3A.

To better understand the genomic evolutionary differences, we further compared the LCBs of SH-01, CyHV-1, and CyHV-3 (Figure 3B). Compared with the genome of SH-01 (290,428 bp), this was increased by 716 bp and 4718 bp in CyHV-1 and CyHV-3, respectively, with 55 and 40 differential genes, including deletions and insertions, and there are 122 homologous genes among the three viruses (Table 1, Figure 2, Appendix A). The comparison of the LCBs revealed that both SH-01 and CyHV-3 contained 10 LCBs, with one lacking in CyHV-1, corresponding to positions 209,037–220,265 in the SH-01 genome (11,229 bp, containing *ORF138* and *ORF139*). Additionally, there are some differences in the alignment order and corresponding length of LCBs among the three viruses. Moreover, the orientation of LCB4 (1911 bp, containing *ORF21*–*23*, corresponding to LCB2 of SH-01) and LCB6 (7421 bp, containing *ORF114*–*115*, *ORF120*–*121*, *ORF123*–*124*, and *ORF126*–*129*, corresponding to LCB5 of SH-01) in CyHV-1, as well as LCB6 (10,781 bp, containing *ORF128*–*133* and *ORF135–136*, corresponding to LCB7 of SH-01) in CyHV-3 are arranged opposite to their counterparts in SH-01 (Figure 3B and Table 2). Interestingly, consistent with Figure 2 and Figure 3A, LCB2 (472 bp, containing *ORF140*, corresponding to LCB6 and LCB8 of SH-01 and CyHV-3, respectively) in CyHV-1 jumped to the interval between LCB1 and LCB3, located at the upstream end of the genome. This suggests a possible molecular mechanism in which there may be potential mutational hotspot sites on both sides of the LCB. Furthermore, a phylogenetic tree was constructed based on the aa sequences of helicase (ORF71). This showed the clear clustering of the newly determined sequences with the previously described CyHV-2 and the closest relationship with ST-J1; it is clustered with CyHV-1 and CyHV-3, separate from AngHV-1 (Figure 3C).

## 4. Discussion and Conclusions

In our previous research, we isolated a new strain, named CyHV-2 SH-01, during outbreaks in crucian carp at a local fish farm near Shanghai, China, and confirmed that goldfish also showed high susceptibility and mortality with symptoms similar to HVHN [27]. Here, we present the complete genome structure and molecular characterization of SH-01. Although CyHV-2 is distributed worldwide, different CyHV-2 isolates have not been comprehensively compared. Our results will provide more background for future research.

Similar to the genome structures of viruses in the genus *Cyprinivirus*, our results showed that the complete genome of SH-01 is 290,428 bp in length with an overall G+C content of around 51.60%, including a U and TR region at both ends. It contains 154 predicted ORFs, in which four ORFs (*ORF5*, *ORF6*, *ORF7*, and *ORF8*) are duplicated in the TRs. Notably, the genome of SH-01 shares 99.60% of its sequence identity with that of ST-J1, and similar to ST-J1, SH-01 also contains a 220 bp inverted repeat region downstream of *ORF25C* and *ORF48*, but it was not observed in SY, SY-C1, and CaHV, nor in CyHV-1 and CyHV-3. Moreover, we found several variations in the SH-01 genome compared to the other seven closely related viruses as per the following discussions.

In herpesvirus, many complete genomic sequences have been determined, and a comparison of the aa sequences of the viral genes reveals that the same set of around 40 genes is commonly found in all herpesviruses’ termed “core” genes, which include most of the essential genes of HSV [48,49]. Similar to herpesvirus, among all sequenced alloherpesviruses, twelve genes (*ORF33*, *ORF46*, *ORF47*, *ORF61*, *ORF71*, *ORF72*, *ORF78*, *ORF79*, *ORF80*, *ORF90*, *ORF92*, and *ORF107*), referred to as core genes, are conserved significantly [26]. Our results found that there are seven core ORFs (*ORF19*, *ORF55*, *ORF72*, *ORF88*, *ORF92*, *ORF93*, and *ORF142*) in the SH-01 genome, of which *ORF72* and *ORF92* are significantly conserved among alloherpesviruses (Figure 1A and Appendix A). Previously, Liu et al. (2018) analyzed the aa sequence homology of the 12 core ORFs (*ORF33*, *ORF46*, *ORF47*, *ORF61*, *ORF71*, *ORF72*, *ORF78*, *ORF79*, *ORF80*, *ORF90*, *ORF92*, and *ORF107*) from CyHV-1, ST-J1, SY-C1, SY, CaHV, CyHV-3, AngHV-1, ranid herpesvirus-1 (RaHV-1), ranid herpesvirus-2 (RaHV-2), and ictalurid herpesvirus-1 (IcHV-1) of the alloherpesviruses family. They then found that four core ORFs (*ORF33*, *ORF79*, *ORF92*, and *ORF107*) share a relatively higher similarity among the ten strains [45]. However, SH-01 has seven core ORFs, and only two (*ORF72* and *ORF92*) are identical to those described above, while the other five core ORFs are different from the ten strains above. This suggests that despite the several variations in SH-01, the two genes (*ORF72* and *ORF92*) related to virus replication and structure are highly conserved in this family and can be used as antigen candidates for vaccines.

Gene expression during the lytic replication of herpesviruses is characterized by a distinct chronological sequence involving three main temporal phases, respectively, IE, E, and L genes. Similar to herpesviruses, Tang et al. (2020) recently identified and screened five IE genes (*ORF54*, *ORF121*, *ORF141*, *ORF147*, and *ORF155*), 34 E genes, and 39 L genes in the CyHV-2 ST-J1 genome using HTS combined with cycloheximide (CHX) and cytarabine (Ara-C) inhibitors. They found that all five IE genes were transcribed within 30 min after infection with CyHV-2; E genes, such as *ORF80*, *ORF89*, and *ORF97*, could be detected at 1 h post-infection (hpi), and most of the other E genes appeared at 1–2 hpi, while L genes such as *ORF7* appeared at 6 hpi, and replication was completed within 8 h [36]. Innovatively, with reference to ST-J1, we have marked five IE, 34 E, and 39 L genes on the genome map (Figure 1A and Appendix A), but it needs to be further identified and confirmed experimentally. The expression patterns are the result of complex interactions between herpesviruses and the cytokines of hosts. After the virus invades the host cell, IE genes initiate transcription immediately, relying on the transcription and translation system of the host to provide transcriptional activating proteins that control the transcription of the E and L genes. The E genes encode proteins that are involved in regulating the physiological state of host cells to facilitate viral DNA replication and metabolism. Subsequently, the L genes, which primarily encode structural proteins, begin to be transcribed, ultimately leading to the assembly and release of infectious virions [35,50,51]. In the future, we can focus on the transcriptional regulatory functions of viral genes during CyHV-2 replication and the mechanisms of interaction with the host.

The prediction of the functional features of proteins encoded by the virus is essential for further understanding of the pathogenic properties and infection mechanism of CyHV-2, and this is undoubtedly indispensable for the development of targeted antiviral drugs or vaccines. In present work, 55 putative proteins encoded by SH-01 are predicted to contain one or more conserved domains (Appendix A). A total of 26 ORFs encoding proteins of SH-01 were predicted to possess an SP, and SPs are mainly short peptides located at the N-terminus of proteins that may serve as potential targets for drugs [52]. Specially, we revealed that six ORFs (*ORF25C*, *ORF34*, *ORF52*, *ORF119*, *ORF127*, and *ORF151A*) encoding proteins contain an SP but no TMD, which indicates that these proteins could be secreted. Moreover, ORF64, ORF114, ORF152A, ORF16, and ORF153B of SH-01 have 10, 9, 8, 7 and 6 TMDs, respectively. These proteins contain many TMDs that span the membrane multiple times, indicating that they may be important membrane proteins for CyHV-2; they may have important functions in substance transport, signal transduction, and membrane receptor recognition and serve as potential antiviral drug targets. Interestingly, latent membrane protein-1 (LMP1) encoded by the Epstein–Barr virus (EBV) contains a domain appertaining to the TNFR family that participates in many signaling pathways of host cells to influence their proliferation and differentiation for demands of virus replication [53]. The TNFR domain of ORF4 predicted in SH-01 may have a similar function. In addition, we also predict that many proteins encoded by SH-01 play a role in viral DNA replication, metabolism, and repair, however, further research on the mechanisms of CyHV-2 replication is needed.

Furthermore, genome sequence comparisons demonstrate that the variations among SH-01 and the other seven strains are evident. The variations are gene recombinations, including deletions, duplications, rearrangements, and horizontal transfers, and/or nucleotide mutations, including base substitutions, insertions, and deletions (Figure 2 and Figure 3A,B). We found that 71 out of 150 ORFs are identical (100%) between SH-01 and ST-J1, and only 23 (of the total 154 ORFs) have no variation in SH-01 compared with ST-J1, SY, and SY-C1 (Appendix A). This suggests that CyHV-2 has co-evolved with its host, and host adaptation has led to genomic diversity among the strains isolated from different hosts [45]. Notably, one group proposed that CyHV-2 could be divided into two Chinese (C) and Japanese (J) genotypes based on differences between the genomes of SY-C1 isolated from China and ST-J1 isolated from Japan, according to their isolation sites [44]. Given the high genomic similarity between SH-01 and ST-J1, SH-01 may be classified as a J genotype.

Moreover, the derived genome sizes (Table 1) are sequences obtained by sequencing and do not precisely equate to the actual sizes of the viruses because the genome of each CyHV contains many tandem direct reiterations of short sequences, often in complex forms containing partial or scattered repeats; repeated sequences are characteristic of most herpesvirus genomes and their lengths are often variable, leading to heterogeneity in genome size [33]. We further compared the LCBs of SH-01, CyHV-1, CyHV-3, and AngHV-1; there are several differences in the number, orientation, alignment order, and corresponding length of the LCBs. This implies that these four viruses contain inserted and deleted genes, and they underwent events involving genes jumping and/or differences in evolution rates in LCBs under long-term host selection pressure, allowing the viruses to occupy more diverse niches [54]. Specifically, compared with CyHVs, AngHV-1 has the most divergent LCBs, and only seven are homologous, suggesting that they diverged earlier in genome evolution, but share a common ancestor and evolved separately in different directions to adapt to environmental stress. Obviously, genome alignment facilitates downstream evolutionary inferences, such as rearrangement history, phylogeny, prediction of ancestral states, and detection of selective pressures, influencing coding sequences and noncoding sequences [55,56].

Summarizing these findings is valuable for future research. Overall, the complete genome sequence and structure of CyHV-2 SH-01 was analyzed and compared with those of CyHVs, AngHV-1, and CaHV. Several variations were found in SH-01, including nucleotide mutations, deletions, and insertions, as well as gene duplications, rearrangements, and horizontal transfers. Notably, the genome of SH-01 isolated from crucian carp shares 99.60% of its identity with that of ST-J1 isolated from goldfish, implying that SH-01 may have originated from goldfish and had been introduced to crucian carp, which confirms our previous work [27]. Our findings provide information to further understand the CyHV-2 genome through sequencing and sequence mining.

## Figures and Tables

**Figure 1 viruses-14-02068-f001:**
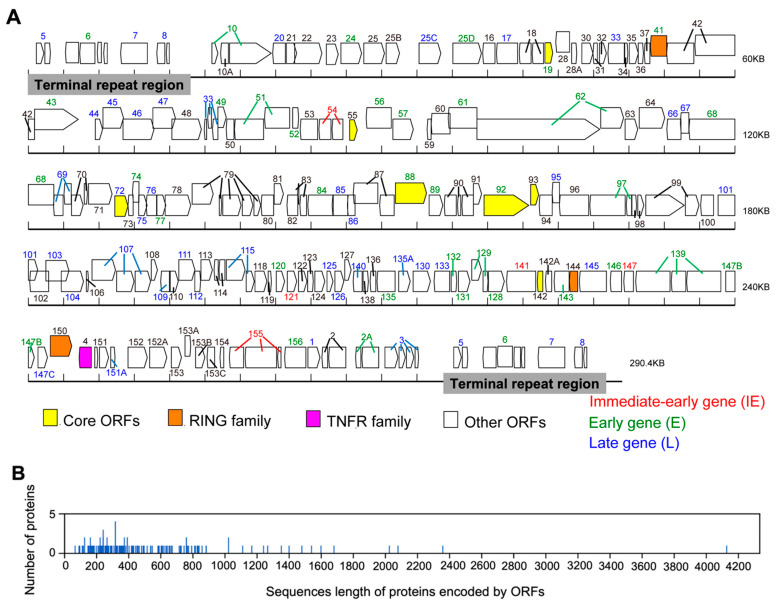
Genome map of CyHV-2 SH-01 and sequence lengths of proteins encoded by open reading frames (ORFs). (**A**) The SH-01 genome is 290,428 bp in length and contains 154 potential ORFs. Arrows indicate the size, location, and orientation of the 154 ORFs, with nomenclature lacking the ORF prefix given below. The number of arrows in an ORF indicates the number of exon-formed coding sequences (CDS) it contains due to the presence of introns. Two terminal repeat (TR) regions are marked with grey boxes. Seven core ORFs are indicated by yellow arrows (Core ORFs). Three ORFs with a RING-finger domain are indicated by orange arrows (RING family). One ORF belonging to the TNFR family is indicated by a pink arrow (TNFR family). The 143 unclassified ORFs are indicated by white arrows (Other ORFs). Five immediate-early (IE), 34 early (E), and 39 late (L) genes are marked in red, green, and blue, respectively, and unidentified genes are marked in black. (**B**) The graph of the sequence lengths of the proteins encoded by 154 ORFs (X-axis) vs. the number of proteins per length was calculated by Geneious Prime v2022.2.1 (Y-axis).

**Figure 2 viruses-14-02068-f002:**
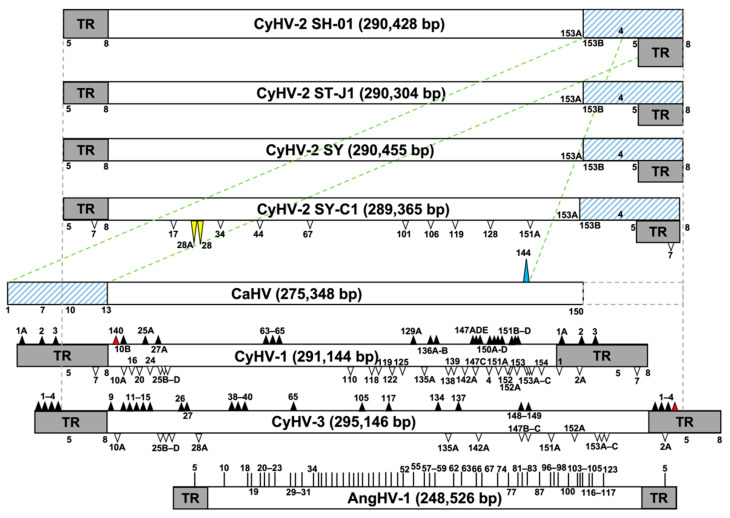
Comparison of genome structures between SH-01 and the other seven strains. Long frames and grey frames indicate the sizes of genomes and terminal repeat (TR) regions, respectively. Long yellow triangles show that the positions of *ORF28* and *ORF28A* in SY-C1 are opposite to each other relative to SH-01 and other isolates of CyHV-2. White triangles in SY-C1, CyHV-1, and CyHV-3 indicate deletions relative to SH-01. Black triangles in CyHV-1 and CyHV-3 indicate insertions relative to SH-01. *ORF140*, with a large translocation in CyHV-1, and *ORF4*, located at the TR in CyHV-3, distinct from SH-01, are marked as red triangles. The dashed frame at the upstream end of the CaHV genome indicates the nucleotides deleted relative to SH-01. The long, green, diagonal dashed lines between SH-01 and CaHV indicate that the 5′-terminal *ORF1*–*13* of CaHV (blue shaded frame) corresponds to the 3′-terminal *ORF153B*–*ORF8* (*ORF144*, with long blue triangles at the downstream end, of CaHV corresponds to *ORF4* of SH-01) of SH-01 (blue shaded frame), similar to ST-J1, SY, and SY-C1 (absence of *ORF7*) marked in blue-shaded frames. The short vertical lines in AngHV-1 indicate the ORFs homologous to SH-01.

**Figure 3 viruses-14-02068-f003:**
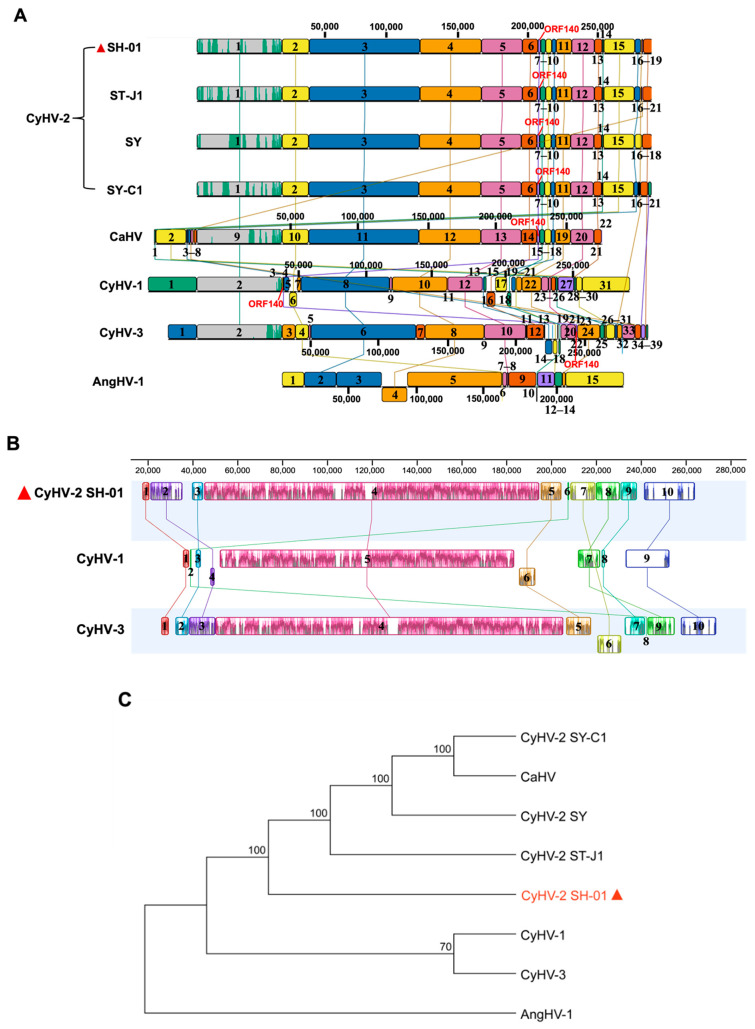
Genomic evolutionary relationships and phylogenetic analysis of SH-01 and the other seven strains. (**A**) Evolutionary patterns among the homologous or heterologous regions of the genomes of eight isolates, including SH-01, were analyzed by Mauve alignment in DNASTAR Lasergene v17.3. Different locally collinear blocks (LCBs) are marked with regions in different colors. Each genome was aligned by the LCB1 of SH-01, and homologous LCBs are connected by corresponding lines. Compared to SH-01, the large “jump” of *ORF140* in CyHV-1 is marked in red. (**B**) Comparison of LCBs among SH-01, CyHV-1, and CyHV-3 using the progressive Mauve algorithm in Geneious Prime v2022.2.1. Different LCBs are marked with regions in different colors, and homologous LCBs are connected by corresponding lines. (**C**) Phylogenetic tree was constructed based on the amino acid sequences of helicase (ORF71) using the neighbor-joining method in MEGA v11 (https://megasoftware.net accessed on 10 July 2022). Bootstrap values of 1000 replications are shown at nodes.

**Table 1 viruses-14-02068-t001:** Genome features of CyHVs, AngHV-1, and CaHV.

Virus		Size (bp)			Nucleotide Composition (%)	No. of ORFs				Identity (%) *^k^*
		Genome	U *^g^*	TR *^h^*	G+C	Genome *^i^*	Unique *^j^*	U *^g^*	TR *^h^*	
CyHV-2	SH-01	290,428	260,586	14,921	51.60	154	150	146	4	***
	ST-J1 *^a^*	290,304	260,238	15,033	51.70	154	150	146	4	99.60
	SY-C1 *^b^*	289,365	259,555	14,905	51.60	143	140	137	3	98.53
	SY *^c^*	290,455	259,749	15,353	51.60	154	150	146	4	98.35
CyHV-1 *^a^*		291,144	224,784	33,180	51.30	143	137	131	6	42.72
CyHV-3 *^d^*		295,146	250,208	22,469	59.20	163	155	147	8	44.54
AngHV-1 *^e^*		248,526	227,258	10,634	53.00	134	129	124	5	36.52
CaHV *^f^*		275,348	-	-	51.73	150	150	-	-	92.63

*^a^* Reprinted with permission from Ref. [33]. 2013, Andrew J. Davison; *^b^* Reprinted with permission from Ref. [44]. 2015, Lijuan Li; *^c^* Adapted with permission from Ref. [45]. 2018, Bo Liu; *^d^* Adapted with permission from Ref. [46]. 2007, Takashi Aoki; *^e^* Adapted with permission from Ref. [24]. 2010, Steven J. van Beurden; *^f^* Adapted with permission from Ref. [26]. 2016, Xiaotao Zeng; *^g^* U represents the unique region in the genome (except in that of CaHV); *^h^* TR represents the terminal repeat in the genome (except in that of CaHV); *^i^* number of ORFs in U plus two copies of TR; *^j^* number of ORFs in U plus one copy of TR; *^k^* the genome identities of eight strains were aligned through MAFFT by Geneious Prime v2022.2.1. CyHV-2 SH-01 (GenBank accession No. BankIt2436221); CyHV-2 ST-J1 (NC_019495.1); CyHV-2 SY-C1 (KM200722.1); CyHV-2 SY (KT387800.1); CyHV-1 (NC_019491.1); CyHV-3 (NC_009127.1); AngHV-1 (NC_013668.3); CaHV (KU199244.1).

**Table 2 viruses-14-02068-t002:** Comparison of LCBs among the SH-01, CyHV-1, and CyHV-3 genomes.

SH-01			CyHV-1			Change (bp) *^b^*	CyHV-3			Change (bp) *^b^*
LCB	Position	Length (bp)	LCB	Position	Length (bp)		LCB	Position	Length (bp)	
LCB1	17,215–20,413	3199	LCB1	35,374–38,196	2823	−376	LCB1	25,754–29,079	3326	+127
LCB2	20,806–35,267	14,462	LCB4	47,773–49,683	1911	−12,551	LCB3	38,157–49,904	11,748	−2714
LCB3	39,346–44,586	5241	LCB3	41,045–43,535	2491	−2750	LCB2	32,129–37,868	5740	+499
LCB4	44,914–195,079	150,166	LCB5	51,890–183,835	131,946	−18,220	LCB4	50,081–205,795	155,715	+5549
LCB5	195,702–204,934	9233	LCB6	185,864–193,284	7421	−1812	LCB5	207,192–218,326	11,135	+1902
LCB6	207,694–208,337	644	LCB2	38,663–39,134	472	−172	LCB8	242,836–243,294	459	−185
LCB7	209,037–220,265	11,229	none *^a^*	none *^a^*	none *^a^*	−11,229	LCB6	221,009–231,789	10,781	−448
LCB8	220,500–231,255	10,756	LCB7	212,442–222,389	9948	–808	LCB9	243,510–255,637	12,128	+1372
LCB9	231,395–238,734	7340	LCB8	222,900–224,492	1593	−5747	LCB7	233,309–242,408	9100	+1760
LCB10	241,885–264,893	23,009	LCB9	233,688–253,332	19,645	−3364	LCB10	258,414–274,200	15,787	−7222

*^a^* “None” responses in which no parallel LCB exists in the genome of CyHV-1; *^b^* the changes in base pairs in the parallel LCBs of the CyHV-1 or CyHV-3 genome relative to SH-01. “+” responses decrease in base pairs, while “−” responses increase in base pairs.

## Data Availability

Data is contained within the article and Appendix A.

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
