# Peer review of "Complete Genome and Molecular Characterization of a New Cyprinid Herpesvirus 2 (CyHV-2) SH-01 Strain Isolated from Cultured Crucian Carp"

_viruses, 2022, doi:10.3390/v14092068_

Round 1
Reviewer 1 Report
Present study reported the complete genome and characters of a new cyprinid herpesvirus 2 strain (CyHV-2 SH-01). Comparison with other cyprinid herpesvirus strains provided new understandings to the genome structure and evolutions of fish herpesviruses. The MS could be considered for publication with minor revisions.
Major issues:
1. It is better to make the fig 1A bigger to make it clearer.
2. Fig 1A, there are several ORF numbers with more than one indicatrix. For example, there are five lines indicating ORF79. It’s not clear so many lines mean. Maybe it indicates the exons.
3. Table 2 and Table 3 can be combined as one table, and comparisons with other strains including CaHV and AngHV-1 could be added to stick out the molecular character of the SH-01.
4. In discussion, the unique character including gene structure / distribution etc and their biological function of SH-01 should be discussed.
5. The authors stated that there are seven core ORFs in the SH-01 genome (line 128-129), but it’s unknown how to define the core ORFs.
Minor issues:
1. The numbers indicate the cited references in the text were not displayed in correct style.
2. line 83, “coursed” should be “caused”.
3. line 112, “Bankit” is not the GenBank accession number.
Author Response
TRANSLATE with x English
| Arabic | Hebrew | Polish |
| Bulgarian | Hindi | Portuguese |
| Catalan | Hmong Daw | Romanian |
| Chinese Simplified | Hungarian | Russian |
| Chinese Traditional | Indonesian | Slovak |
| Czech | Italian | Slovenian |
| Danish | Japanese | Spanish |
| Dutch | Klingon | Swedish |
| English | Korean | Thai |
| Estonian | Latvian | Turkish |
| Finnish | Lithuanian | Ukrainian |
| French | Malay | Urdu |
| German | Maltese | Vietnamese |
| Greek | Norwegian | Welsh |
| Haitian Creole | Persian |
TRANSLATE with EMBED THE SNIPPET BELOW IN YOUR SITE Enable collaborative features and customize widget: Bing Webmaster Portal Back

Reviewer 2 Report
In the article "Complete genome and molecular characterization of a new Cyprinid Herpesvirus 2 (CyHV-2) SH-01 strain isolated from cultured crucian carp " the authors describe and compare genomes of CyHV-2 with CaHV, CyHV-1, CyHV-3, and also AngHV-1. The article is written in an understandable way. The introduction provides necessary information about viruses and informs what will be presented in this article, but in the text, reference numbers should be placed in square brackets [ ], and placed before the punctuation; for example [1], [1–3] or [1,3]. Correct it in the whole article. Please add also some information about CaHV in the “Introduction” part and explain the shortcut.
The material and methods part describes virus isolation and obtained results by WGS. I think this part is too short. There is no information about preparing samples to sequencing. Obtained results from 2.2 point should be in the “Results” part.
All tables and figures are readable and put in the right place in the text. Results are presented very clearly and understandable. The discussion refers to other references and refers to obtained results.
I think this is a valuable article which should be published. The authors presented their research in an understandable way. They put a lot of work into their research.
Author Response

(The authors gave the same response as above.)

Reviewer 3 Report
In this article, the complete genome of a new Cyprinid herpesvirus 2 (CyHV-2) SH-01 strain isolated from diseased crucian carp were sequenced and analyzed, the function features of proteins and chronological characteristics of gene expression were predicted, and the genome structure features and genomic evolutionary relationships were compared with another closely related viruses. The results provided new clues to better understand the CyHV-2 genome through sequence mining. This study is worthy of publication in Viruses after some issues are resolved.
1. The initial letter of the second word of “Cyprinid Herpesvirus 2” in the title should use lowercase letter “h”, also in Line 22 “Crucian”. This type of format should be standardized throughout the text.
2. Line 30: In abstract, the description of “99.98% homology” is improper, “identity” is used in the text for the genomic sequence comparison analysis of different strains, in addition, “homology” shouldn’t describe by degree and different sequences can only be homologous or heterologous.
3. The reference numbers in the main text lack the brackets in PDF file as required by the Viruses format, please verify and add them.
4. Page 2: In introduction, Line 49-51, in the description of the spread of CyHV-2 to several countries and regions worldwide, the citation of the USA, Italy, the Netherlands and China should use the earliest reported references, these should be verified. Moreover, the reports of CyHV-2 infection in China (citing the earliest reported reference) should also be listed together rather than described separately, otherwise it may cause ambiguity. So, these sentences need to be reorganized.
Line 48: “spread into” changed into “spread to”
5. In Results, line 95-98 “A total of 46,723,798 95 raw reads (7,008,569,700 raw bases) were obtained……with a G+C content of 59.5%”ï¼› line 103-106 “one contig was obtained with a length of 290,428 bp…… which confirmed that the genome belongs to CyHV-2”; line 110-112 “We successfully assembled the complete genome sequence of CyHV-2 SH-01 and submitted it to GenBank (Accession No. Bankit2436221).” All these are the results of this study and should be removed to the Results section.
In addition, more details should be provided in Materials and Methods, such as, descriptions for comparation with CyHV-1 and CyHV-3, and for genomic evolutionary relationships analysis.
6. line 148-154, the description about the findings of Tang et al [2020] are not the results of this study, they should be removed to the Discussions.
7. In Results, line 118 and line 161-166, the website link should be removed to Material and Methods.
Author Response

(The authors gave the same response as above.)
